# Indirect COVID-19 health effects and potential mitigating interventions: Cost-effectiveness framework

Sigal Maya[1]*, James G. Kahn[1,2], Tracy K. Lin[3], Laurie M. Jacobs[1], Laura A. Schmidt[1,4], William B. Burrough[5], Rezvaneh Ghasemzadeh[5], Leyla Mousli[1], Matthew Allan[1], Maya Donovan[1], Erin Barker[1], Hacsi Horvath[1], Joanne Spetz[1], Claire D. Brindis[1,6], Mohsen Malekinejad[1,2]

1 Philip R. Lee Institute for Health Policy Studies, University of California San Francisco, San Francisco, CA, United States of America, 2 Department of Epidemiology and Biostatistics, University of California San Francisco, San Francisco, CA, United States of America, 3 Institute for Health and Aging, University of California San Francisco, San Francisco, CA, United States of America, 4 Department of Humanities and Social Sciences, University of California San Francisco, San Francisco, CA, United States of America, 5 University of California San Francisco Benioff Children's Hospital Oakland, Oakland, CA, United States of America, 6 Department of Pediatrics, University of California San Francisco, San Francisco, CA, United States of America

* Sigal.Maya@ucsf.edu

**Data Availability Statement:** All relevant data are within the article and its Supporting Information files.

## Abstract

### Background

The COVID-19 pandemic led to important indirect health and social harms in addition to deaths and morbidity due to SARS-CoV-2 infection. These indirect impacts, such as increased depression and substance abuse, can have persistent effects over the life course. Estimated health and cost outcomes of such conditions and mitigation strategies may guide public health responses.

### Methods

We developed a cost-effectiveness framework to evaluate societal costs and quality-adjusted life years (QALYs) lost due to six health-related indirect effects of COVID-19 in California. Short- and long-term outcomes were evaluated for the adult population. We identified one evidence-based mitigation strategy for each condition and estimated QALYs gained, intervention costs, and savings from averted health-related harms. Model data were derived from literature review, public data, and expert opinion.

### Results

Pandemic-associated increases in prevalence across these six conditions were estimated to lead to over 192,000 QALYs lost and to approach $7 billion in societal costs per million population over the life course of adults. The greatest costs and QALYs lost per million adults were due to adult depression. All mitigation strategies assessed saved both QALYs and costs, with five strategies achieving savings within one year. The greatest net savings

**Funding:** Authors would like to thank the Stupski Foundation (https://stupski.org/) for their financial contributions that made the completion of the RAPID project possible. Several authors of this manuscript received supplemental funding from other funders to complete the manuscript: SM and JGK received funding from National Institute on Drug Abuse (https://nida.nih.gov/) #5R37DA015612-170 and LAS and JS from #R01DA047379, LAS received funding from Center for Advancing Translational Sciences at the National Institutes of Health (https://ncats.nih.gov/) #UL1 TR001872, and MM received funding from California Health Care Foundation (https://www.chcf.org/). CDB's time was supported in part by the Health Resources and Services Administration (HRSA, https://www.hrsa.gov/) of the U.S. Department of Health and Human Services (HHS) (under #U45MC27709, Adolescent and Young Adult Health Capacity Building Program, and from HRSA grant # MCHB 5T71MC00003, Leadership education in Adolescent Health). This information or content and conclusions are those of the authors and should not be construed as the official position or policy of, nor should any endorsements be inferred by HRSA, HHS, the U.S. Government or any other funder. The sponsors had no role in the preparation or submission of this article.

**Competing interests:** The authors have declared that no competing interests exist.

over 10 years would be achieved by addressing depression ($242 million) and excessive alcohol use ($107 million).

## Discussion

The COVID-19 pandemic is leading to significant human suffering and societal costs due to its indirect effects. Policymakers have an opportunity to reduce societal costs and health harms by implementing mitigation strategies.

## Introduction

Over the first 18 months of the SARS-CoV-2 (COVID-19) pandemic, public health efforts have focused on reducing direct health impacts of the virus by preventing and clinically managing infection. However, much evidence reveals profound indirect mental and physical health impacts. The California population has experienced a wide range of stressors, such as a prolonged period of fear, grief, and uncertainty, social isolation, and economic dislocation and hardship [1–3]. Such stressors can activate the biological stress response, leading to short- and long-term health effects [4].

Studies have shown significant increases in the prevalence of varied health harms resulting from the COVID-19 pandemic in the US and California. Depression [5,6], anxiety [5,6], family violence [7–10] and excessive alcohol and substance use [5,11,12] have increased. Poorer health outcomes, such as increased stroke mortality [13–15], higher prevalence of out-of-hospital cardiac arrests [16], and delays in cancer care and treatment [17] have also been observed [18]. As the COVID-19 pandemic evolves, government efforts will increasingly shift towards responses to indirect health harms and related high costs which could persist for many years.

The dynamic nature of the pandemic and California's unfolding response requires decisionmakers to anticipate which indirect health impacts will prove especially costly over time and which cost-effective interventions could mitigate adverse health and financial outcomes. An effective, comprehensive policy response requires understanding the indirect impacts of the pandemic, their short- and long-term economic and health harms, and the likely effects and costs of potential mitigating interventions. To provide this information for the State of California, we formed an interdisciplinary rapid-response scientific taskforce and policy simulation laboratory. Our study, entitled RAPID (Rapid Assessment of Pandemic Indirect Impacts and Mitigating Interventions for Decision-making), aimed to systematically evaluate and analyze the pandemic's impact on an important subset of indirect health outcomes and to estimate the effect of evidence-based mitigation strategies in ameliorating health and cost outcomes.

## Methods

### Overview

We developed a health and cost simulation tool employing a generic structure to portray the health and cost effects of COVID-19 pandemic-associated increases in prevalence of individual conditions. The model, called "Broad & Rapid Analysis of COVID-19 Indirect Effects" (BRACE), uses a cost-effectiveness framework that can portray short- and long-term health and cost effects and the impacts of mitigating interventions (e.g., rent support or counseling) on health and societal costs for any condition exacerbated by the pandemic (e.g., homelessness, depression). Health outcomes include deaths and quality-adjusted life years (QALYs). Costs

include both direct medical and non-medical costs. The model specifies the coverage of mitigating interventions. Sensitivity analyses estimate how model input uncertainties (singly and collectively) affect results. Data were derived from literature review, public data (e.g., government reports), and expert opinion. Models were built in Excel (v16, Microsoft 365), and sensitivity analyses were conducted with @RISK (v8.2, Palisade Corporation) software.

## Model structure

We modeled each indirect health condition separately. Model parameters are included in S1 File, and the functioning BRACE workbook is available on request.

Each condition's model is split into two sections: (1) added burden due to the pandemic and (2) intervention outcomes. The added burden section incorporates condition prevalence before the pandemic, short- and long-term costs and health state utility loss and mortality per episode, and the risk ratio for the condition during the pandemic. The model estimates total and excess prevalence of the condition during the COVID-19 pandemic, and excess direct costs and QALYs lost per capita. The intervention section takes the outputs from the added burden section, along with intervention effectiveness and cost, to estimate net costs (or savings) and QALYs gained with the intervention per person and scaled up to 20% of eligibles.

## Selection of health conditions and mitigating interventions

Using a systematic approach, we searched peer-reviewed literature to identify six high-priority indirect public health conditions that increased during the pandemic (Table 1): depressive symptoms, excessive alcohol use, opioid use disorder (OUD), homelessness, intimate partner violence (IPV), and stroke mortality. Selection criteria included availability of robust research as of November 2020, although many other conditions are likely to be affected by the pandemic. See S2 File for detail on search strategy and strength of evidence assessment.

For each health condition, we first identified commonly-used, evidence-based interventions that were recommended by relevant health organizations (e.g., SAMHSA website, US Surgeon General reports, US Veterans Administration/Department of Defense treatment guidelines) and content experts. Next, we examined the peer-reviewed literature on each intervention, prioritizing systematic reviews and meta-analyses, to confirm that each mitigation strategy had evidence to support its effectiveness and cost-effectiveness at the population level. Our goal was to understand where gains could be achieved across important health conditions (rather than comparing cost-effectiveness of different interventions for each condition). Therefore, we extracted data on one prevention or treatment intervention per condition that was relevant for

**Table 1. The human toll: Increased health harms due to the COVID-19 pandemic.**

| Priority Public Health Condition Indirectly Affected by COVID-19 | Relative Risk: Change Under COVID-19 (uncertainty range) | Excess Quality-Adjusted Life Years (QALYs) Lost Due to COVID-19 per Million Total Population | |
|---|---|---|---|
| | | **Short-Term** | **Long-Term** |
| **Depressive symptoms [19–21]** | 1.37 (1.20–2.56) | 24,831 | 43,807 |
| **Intimate partner violence [22–24]** | 1.11 (1.05–1.16) | 2,445 | 49,902 |
| **Homelessness [25]** | 6.67 (5.34–8.00) | 12,951 | 3,527 |
| **Excessive alcohol use [12,20]** | 1.19 (1.07–1.42) | 5,928 | 19,807 |
| **Opioid use disorder [26,27]** | 1.63 (1.33–1.98) | 3,239 | 24,877 |
| **Stroke mortality [13–15,28–30]** | 1.53 (1.40–1.67) | 1,214 | -- |

QALYs: Quality-adjusted life years.

California, had higher strength of evidence, and for which effect size estimates were reported in or could be transformed into a risk ratio for dichotomous clinical outcomes (and thus were compatible with BRACE). We also considered constraints on rolling out programs under pandemic conditions, including sheltering-in-place and whether individuals would likely avail themselves of services, as our main exclusion criterion (S3 File). The final mitigation strategies included: Cognitive-behavioral therapy (CBT) and antidepressants for depressive symptoms, nurse-family partnership for IPV, rent subsidies for homelessness, screening and brief intervention for excessive alcohol use, medication-assisted treatment (MAT) for OUD, and public awareness campaign for stroke mortality.

For all mitigation strategies except the public awareness campaign for reducing stroke mortality, we assumed that the intervention's reach was 20% of adults affected by each condition (regardless of whether the individuals are experiencing the condition due to COVID-19), scaled to a total general population of one million adults. This target was arbitrary, intended to be ambitious yet realistic. (As a point of reference, while 66% of depression is treated in the US, under 20% of those with OUD receive treatment [31,32]). For example, for a population of one million adults, if the prevalence of a condition was 2% (i.e., 20,000 affected adults), we assumed the intervention reached 20% of them (i.e., 4,000 adults). For the public awareness campaign to reduce stroke mortality, we assumed 100% potential reach among those at risk for in-hospital stroke mortality and adjusted the cost of intervention to reflect the cost per person in this population.

### Health outcomes

Each model included the prevalence of the condition before the COVID-19 pandemic, derived from epidemiologic studies, and estimates of the effect of the COVID-19 pandemic on the condition, based on the risk ratio derived from the literature review. For homelessness, recent data were unavailable to assess the change during COVID-19 because the predicted homelessness had not yet occurred given eviction moratoria in California. Thus we used data from the Great Recession of the late 2000s as a proxy, given its comparable economic impact.

For each condition, we estimated health outcomes (deaths and disability) per episode using data identified in literature. Morbidity was estimated using health state utility decrements associated with each condition [33], while deaths were calculated using case-fatality ratios. Of note, we assumed that pandemic-related homelessness would have markedly lower consequences than chronic homelessness (e.g., acute, likely to stay with friends/family rather than shelter/street), so we reduced literature-derived health outcome values by 75%. For all conditions, we identified values for short- and long-term (often lifetime) outcomes. Long-term health outcomes were assumed to occur after a mean of 10 years and discounted to the present using a 3% annual discount rate. Health outcomes were standardized into QALYs, which combine morbidity (decreased health state utility for specified durations) and mortality (likelihood and associated QALY loss), to allow comparison across different diseases and interventions.

We then measured changes in short- and long-term health outcomes that may result from implementing specific mitigating strategies. This was achieved by applying reported intervention efficacy (relative reduction in the condition) to the condition prevalence during COVID-19.

### Costs

Costs were considered from a societal perspective, rather than costs to specific actors (e.g., the government or insurers). The primary sources for cost data were peer-reviewed publications and grey literature, which often provided California-specific cost information. When published cost data were unavailable, we relied on payment rates from the Medi-Cal fee schedule

updated monthly by the California Department of Health Care Services. Costs were adjusted for inflation and presented in 2020 US dollars. Condition-specific cost estimates included short- and long-term medical costs (e.g., ambulance, emergency room visits, medicine) and direct non-medical costs (e.g., police intervention) per episode. (Electronic supplement for costing details upon request.) Short-term costs were assumed to be incurred in the first year and not discounted. Long-term costs were assumed to be incurred after a mean of 10 years and discounted accordingly.

We estimated mitigation strategy costs following the same approach as above. The interventions we examined are typically limited in duration, with costs entirely or predominantly incurred during year one, and thus they were not discounted.

We calculated net savings over the course of an individual's lifetime, and at one, three, and ten years following intervention implementation. We assumed that 60% of short-term costs and none of the long-term costs were incurred in year one. Over three years, 90% of the short-term costs and 20% of long-term costs were incurred. After year ten, 95% of short-term costs and 50% of long-term costs were incurred.

### Cost-effectiveness

If the intervention cost exceeded savings over the lifetime, we calculated a cost-effectiveness ratio (net cost per QALY gained). If the intervention cost was less than the savings, the intervention was described as "dominant," meaning that it both cost less than doing nothing (due to savings from averted clinical and other outcomes) and improved health outcomes. Per-person estimates were scaled up to calculate lifetime anticipated costs, savings, and QALYs gained per general adult population of one million.

## Results

We present results pertaining to overall health and economic costs for each condition, and how these could change as the result of mitigation for indirect health outcomes of the pandemic. Detailed findings for individual conditions are in S1 File.

### Indirect health harms of the COVID-19 pandemic

Results suggest wide-ranging indirect health consequences of the pandemic (Table 1). While there were increases of 11% in the rate of IPV by April 2020 compared to before the shelter-in-place order, the prevalence of homelessness is expected to increase nearly sevenfold due the pandemic.

Across the six conditions assessed, we estimated that total QALYs lost were 192,530 per million population. The greatest QALYs per million population were lost to depression (68,638), followed by IPV (52,347). In-hospital stroke mortality increased 53% during COVID-19 compared to before; however, the overall reduction in QALYs (1,214) was not as pronounced as for the other conditions.

### Economic consequences of the indirect health harms of the COVID-19 pandemic

When no mitigation strategies were deploted, the highest costs (both short- and long-term) were estimated to result from failing to mitigate rising rates of depressive symptoms among adults (Table 2). While the cost-per-episode of depressive symptoms from an individual patient perspective was small, the overall medical and non-medical cost rose to $2.2 billion per

**Table 2. Costs of doing nothing: Estimated societal costs from indirect health harms of the COVID-19 pandemic.**

| Priority Public Health Condition Indirectly Affected by COVID-19 | Cost per Episode | Excess Societal Costs Due to COVID-19 per Million Total Population | |
|---|---|---|---|
| | | Short-Term | Long-Term |
| Depressive symptoms [34,35] | $32,599 | $954M | $1,277M |
| Intimate partner violence [36] | $115,562 | $373M | $326M |
| Homelessness [37,38] | $99,969 | $706M | $1,448M |
| Excessive alcohol use [39–43] | $94,004 | $138M | $952M |
| Opioid use disorder [44–47] | $79,551 | $166M | $385M |
| Stroke mortality [48] | $16,773 | $844,000 | -- |

M: Million.

million total population. The magnitude of these costs was largely due to the sheer extent of the excess prevalence of depressive symptoms during the pandemic.

Other public health problems, such as homelessness, were also likely to incur large medical and non-medical costs. Even though homelessness had a lower QALY burden than other conditions, it could impose a large economic burden (nearly $2.2 billion per million total population) if not fully addressed. Increased rates of excessive drinking during the pandemic were also projected to cost almost $1.1 billion per million total population.

## Effect of mitigating interventions

All intervention outcomes assumed 20% reach of the affected population, except for the public awareness campaign for stroke mortality with 100% reach. All interventions produced medical and non-medical cost-savings that increased over time, while also averting losses in QALYs (Table 3). All but one intervention, MAT for OUD, was projected to yield net cost savings within three years. By ten years, all interventions were expected to provide net cost savings.

The largest estimated health gains were from intervening in IPV, with over 22,000 QALYs gained per million adult population (Table 3). This was followed by treating depressive

**Table 3. Mitigation strategies: Health and economic outcomes from intervention programs that reach 20%[*] of affected adults per million total population.**

| Priority Public Health Condition Indirectly Affected by COVID-19 and Associated Interventions | QALYs Gained | Intervention Costs | Cost-Effectiveness vs. Doing Nothing (lifetime) | Net Savings by Time Period | | |
|---|---|---|---|---|---|---|
| | | | | Over 1 Year | Over 3 Years | Over 10 Years |
| **Depressive symptoms** | | | | | | |
| Cognitive-behavioral therapy +antidepressants | 12,707 | $44.5M | Dominant | $61.5M | $161.7M | $241.5M |
| **Intimate partner violence** | | | | | | |
| Nurse-family partnership | 22,186 | $164.7M | Dominant | ($69.8M) | $5.3M | $54.6M |
| **Homelessness** | | | | | | |
| Rent subsidies | 1,648 | $47.1M | Dominant | ($4.7M) | $45.4M | $92.4M |
| **Excessive alcohol use** | | | | | | |
| Screening and brief intervention | 4,835 | $6.6M | Dominant | $8.9M | $52.5M | $107.4M |
| **Opioid use disorder** | | | | | | |
| Medication-assisted treatment | 5,674 | $56.2M | Dominant | ($36.0M) | ($10.5M) | $14.5M |
| **Stroke mortality** | | | | | | |
| Public awareness campaign | 388 | $14,350 | Dominant | $147,500 | $228,500 | $242,000 |

[*]Except for stroke, which assumes 100% coverage of those at risk for stroke. QALYs: Quality-adjusted life years, M: Million. Parentheticals indicate net costs.

symptoms, which led to nearly 13,000 QALYs gained. Intervening to prevent stroke mortality led to smaller gains (about 400 QALYs) due to the relative rarity of the condition and the modest effects of a public awareness campaign; this was still a dominant strategy with net savings.

Greatest savings came from providing CBT and antidepressants to people experiencing depressive symptoms. Within one year, $61.5 million net savings were estimated with this intervention. Screening, brief intervention, and referral to treatment (SBIRT) for individuals experiencing increased alcohol use was estimated to save $8.9 million within one year.

## Sensitivity analyses

Probabilistic sensitivity analyses indicated that interventions reliably saved both QALYs and costs across a range of uncertainties in key inputs (Fig 1). The greatest gains in both QALYs and dollars per million adult population were attained with either CBT and antidepressents for depressive symptom mitigation, which resulted in mean net savings of $383 million (95% prediction interval (PI): $130 million–$786 million) and mean QALYs gained were 13,136 (95% PI: 5,632–24,105). Alternatively, the nurse-family partnership program could save mean 21,494 QALYs (95% PI: 5,664–37,146) by mitigating the effects of IPV, but with lower overall net savings of $121 million (95% PI: -$99 million–$358 million). Among the remaining four interventions, SBIRT for excessive alcohol use could yield comparatively greater dollar savings ($199 million, 95% PI: $85 million–$367 million). MAT for OUD was associated with a mean 5,933 QALYs gained (95% PI: 2.887–9,943) and $59 million in net savings (95% PI: $10 million–$130 million).

Due to high uncertainty in the increase in prevalence of homelessness, we conducted a one-way deterministic sensitivity analysis on this parameter. If the relative risk for homelessness during the pandemic was one-fourth of the base case value (1.67 versus 6.67), QALYs gained were reduced to 369, and net savings to $14 million. Rent subsidies remained dominant, although savings did not occur until after three years.

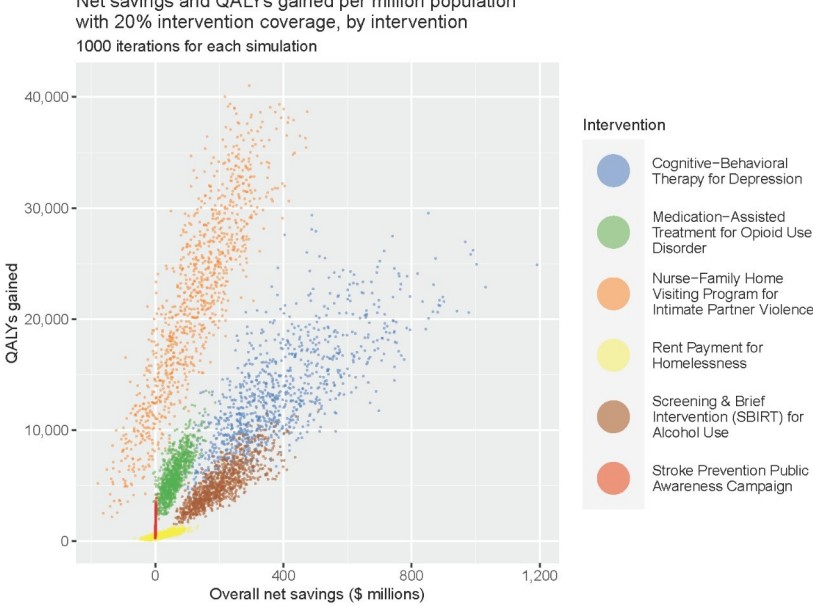

**Fig 1. Comparison of health benefits and economic savings of alternative mitigation strategies.** QALY: Quality-adjusted life year. 20% intervention coverage for all interventions except stroke prevention public awareness campaign, which assumes 100% coverage.

## Discussion

Policymakers need scientific guidance to craft interventions that address the long-term indirect effects of the COVID-19 pandemic. The BRACE model provides a flexible tool for assessing the health and economic impacts of these indirect effects. We found that adverse conditions increased at widely vaying rates during the pandemic, ranging from an 11% increase in IPV to a 63% increase in OUD. The rate of in-hospital stroke mortality increased over 50%, likely due to delayed presentation to care settings due to fear of contracting SARS-CoV-2 or delays in receiving definitive treatment once in the hospital due to additional precautionary measures [13–15,30]. While increases in homelessness were not actually observed, given the temporary nature of eviction moratoria (which were issued for a few months at a time with no certainty of extension), we nevertheless included this condition as the 700% potential rise in homelessness was worrisome. The health harms and economic tolls of all these changes were substantial, with an estimated 192,000 QALYs lost and nearly $7 billion net cost per million population, primarily due to increased depressive symptoms. Mitigating these harms now will achieve significant cost-savings within one-to-three years. Across the six interventions considered, an estimated 47,000 QALYs were gained and potential net savings reached $5 billion, for a population of one million adults.

Some health and cost burdens have already been incurred. However, the sooner mitigation strategies are implemented, the more suffering and long-term costs can be averted by arresting increases in the severity of indirect health conditions and limiting downstream impacts. While these conditions may eventually return to pre-pandemic levels, interventions described in this study would still have health benefits and cost-savings, since they improve health for both pandemic-related and pre-existing cases. The magnitude of benefits will depend on the extent that indirect health impacts continue. Additionally, intervention benefits may vary for different populations within California. Due to the novelty of the pandemic, data on indirect effects for different populations (e.g., by geographic location, race/ethnicity, economic status) were not available at the time of analysis, thus we did not attempt to evaluate differences in intervention outcomes across groups. However, we believe results would be similar even in populations for which interventions may be less effective, given our consistently strong favorable findings.

The findings of this analysis were more favorable than published studies for the individual conditions [49–51]. Prior studies often include only short-term costs and are generally limited to direct medical costs, leading to smaller estimates of savings from mitigation. We were also more likely than others to find dominance because our comparator was 'no intervention'; we did not evaluate the incremental benefit of a more expensive intervention over a less expensive one. We also did not incorporate productivity losses or the spillover effect of the evaluated conditions on children, which may lead to additional savings.

We described a menu of economically attractive alternatives for mitigation following the first year of the COVID-19 pandemic. Other considerations apply when implementing these interventions, including community acceptability and barriers to implementation. For example, interventions that require in-person contact can be challenging if specific populations are not confident about seeking healthcare services. Decisionmakers must be attentive to these potential barriers and identify locally-appropriate responses. Moreover, the changing pandemic context requires agility in the face of evolving impacts to prevent human suffering and financial costs. Specifically, the effects of the pandemic on children are likely to be substantial and long-lasting [52,53].

Governments have access to a variety of potential interventions that may ameliorate the adverse health events resulting from the COVID-19 pandemic. These interventions include financing the expansion of existing clinical interventions (e.g., counseling and medications for

depression), leveraging *macro-level economic policies* (e.g., financial stimulus packages; interest rate reductions; rent, eviction, and mortgage moratoria; relaxed conditions of debt payments and income taxes), and using *organization-level measures* (e.g., school enhancements, workplace safety protections) and *community public health interventions* (e.g., information campaigns). When considering results from the BRACE model, which estimated the cost and effect of each condition separately, one might consider leveraging upstream mitigation strategies that may generate benefits across multiple conditions. Such strategies include economic tools (e.g., earned income tax credit, enhanced unemployment benefits) that have unique advantages in the current context; they tend to reach a broad swath of the population, can mitigate multiple negative health outcomes, and may have lower administration burdens. These strategies are particularly valuable for building population resistance to stressors on health that unfold over time (e.g., exposure to family violence, depression). Insights from the BRACE model should be incorporated and synchronized with existing programs and systems of care, to leverage existing infrastructure, reduce redundancy, and potentially lead to large health impacts and cost savings if streamlined and coordinated.

In addition to ongoing planning and research that builds from BRACE, we recommend establishing a tracking and evaluation system with built-in quality improvement strategies to ascertain how interventions are being implemented. These data could help refine and improve interventions to assure that those needing support do indeed receive it. As always, it is important to demonstrate concrete returns on investment for preventive health services, which, while typically quite large, can easily become taken-for-granted. Process and outcome studies could also be conducted, using qualitative and quantitative methods and community-engaged research approaches, to measure short- and longer-term outcomes, best practices, adaptations, and anticipated and unanticipated outcomes. An equally important action is compilation of cost data, including information on who pays for the intervention and who reaps benefits.

## Limitations

Our simulation tool enables rapid assessments of health conditions and evidence-based interventions. Favoring the ability to consider multiple health conditions simultaneously with adequate precision for policy decisions required omitting the usual disease-specific nuance used to portray clinical progression and multiple interventions. Detailed estimation of costs and health outcomes was also beyond the scope of this analysis. Further, we considered each intervention individually, disregarding any potential cost-sharing. Greater savings could be achieved by streamlining interventions and leveraging synergistic effects. More sophisticated modeling could be linked with the model as urgency, resources, and priority allow. We suspect that the broad conclusions would remain unchanged.

We were limited by availability of robust data at the time of analysis and by the model requirement of dichotomous efficacy data, which restricted the selection of health conditions and interventions. While we sought guidance from content area experts, the timeline and breadth of the project inhibited more intensive formal approaches (e.g., Delphi elicitation) that might have been ideal. Precise efficacy of interventions (especially within the context of the pandemic) remained uncertain; different strategies may lead to greater health gains and/or cost savings. Our assumption of 20% intervention coverage may be overestimating the pool of eligible individuals, thereby overstating costs. Incremental coverage is largely a public health policy decision, although constrained by available capacity and resources. Smaller efforts would result in smaller gains, but identical patterns of net savings, thus not affecting cost-effectiveness ratios. Regardless, this assumption may need to be adjusted depending on available human resources and scale-up processes.

## Conclusions

The COVID-19 pandemic has severely damaged public health around the world. There is an opportunity to ameliorate the indirect health and cost impacts of the pandemic. Evidence-based research and modeling are important tools to inform solutions to these widespread and severe effects. Implementing evidence-based interventions should be a policy priority, particularly as the pandemic has exacerbated inequality in a wide range of socio-economic conditions. We identified and evaluated the impact of mitigation strategies for health conditions that worsened under the COVID-19 pandemic and present a menu of options for improving the health of communities and generate cost-savings. Implementing any of these strategies may lead to significant dollar savings for the State of California and to substantial improvements in the health of millions of state residents, for years to come.

## Supporting information

**S1 File. Model inputs and results.**
(DOCX)

**S2 File. Strength of evidence.**
(DOCX)

**S3 File. Selection of conditions and interventions.**
(DOCX)

## Acknowledgments

We sincerely thank Dr. Nadine Burke Harris, Matt Schueller, Dr. Devika Bhushan, and Dr. Steve Wirtz of the Office of the California Surgeon General for their vision and valuable contributions to this project. We are grateful to Drs. Ralph Catalano, William H. Dow, Jenny Liu, Elliot Marseille, Brigid McCaw, and Justin White for their input and valuable review of previous versions of this work. We also thank Drs. Kirsten Bibbins-Domingo, Jim Lightwood, Wendy Max, Travis Porco, and George Rutherford for sharing their insights as the RAPID project was being conceptualized.

## Author Contributions

**Conceptualization:** James G. Kahn, Laura A. Schmidt, Joanne Spetz, Claire D. Brindis, Mohsen Malekinejad.

**Data curation:** Sigal Maya, Tracy K. Lin, Laurie M. Jacobs, William B. Burrough, Rezvaneh Ghasemzadeh, Matthew Allan, Maya Donovan, Joanne Spetz.

**Formal analysis:** Sigal Maya, James G. Kahn.

**Funding acquisition:** Laura A. Schmidt, Joanne Spetz, Claire D. Brindis, Mohsen Malekinejad.

**Investigation:** James G. Kahn, Laura A. Schmidt, Joanne Spetz, Mohsen Malekinejad.

**Methodology:** Sigal Maya, James G. Kahn, Tracy K. Lin, Hacsi Horvath.

**Project administration:** Leyla Mousli, Erin Barker.

**Software:** Claire D. Brindis.

**Supervision:** James G. Kahn, Laura A. Schmidt, Joanne Spetz, Claire D. Brindis, Mohsen Malekinejad.

**Visualization:** Sigal Maya, Tracy K. Lin.

**Writing – original draft:** Sigal Maya, Tracy K. Lin.

**Writing – review & editing:** Sigal Maya, James G. Kahn, Tracy K. Lin, Laurie M. Jacobs, Laura A. Schmidt, Joanne Spetz, Claire D. Brindis, Mohsen Malekinejad.

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
