## [Decision Letter · Decision Letter 0]

30 May 2022

PONE-D-22-09397Indirect COVID-19 Health Effects and Potential Mitigating Interventions: Cost-effectiveness frameworkPLOS ONE

Dear Ms Sigal Maya,

Thank you for submitting your manuscript to PLOS ONE. After careful consideration, we feel that it has merit but does not fully meet PLOS ONE’s publication criteria as it currently stands. Therefore, we invite you to submit a revised version of the manuscript that addresses the points raised during the review process.

We look forward to receiving your revised manuscript.

Kind regards,

Sebsibe Tadesse, PhD

Academic Editor

PLOS ONE

**Comments to the Author**

1. Is the manuscript technically sound, and do the data support the conclusions?

Reviewer #1: Yes

Reviewer #2: Partly

2. Has the statistical analysis been performed appropriately and rigorously? 

Reviewer #1: Yes

Reviewer #2: No

3. Have the authors made all data underlying the findings in their manuscript fully available?

Reviewer #1: Yes

Reviewer #2: No

4. Is the manuscript presented in an intelligible fashion and written in standard English?

Reviewer #1: Yes

Reviewer #2: Yes

5. Review Comments to the Author

Reviewer #1: This interesting paper presents data on the cost-effectiveness of key evidence-based mitigation strategies that address six priority health conditions that have been indirectly affected by the COVID-19 pandemic in California. It is well written and results are presented clearly and conclusions are drawn from data presented.

As COVID-19 vaccine population coverage continues to increase in many countries, many governments are keen to ease both health and border restrictions to enable economic rebound and communities to recover under ‘COVID-normal’ policies. However, as experts predicted the COVID-19 restrictions and acute focus on managing the COVID-19 pandemic meant that many other health conditions have been deprioritized in the process and over the next 1-5 years, we are likely to see a second wave of health issues that have resulted from this deprioritization.

The authors undertook a systematic review of the recent literature to identify key health conditions that have been exacerbated during COVID pandemic - and findings of the review and selection of the six conditions are presented in supplementary files.

I only have a few minor suggestions:

- Table 1. The Human Toll: Increased Health Harms due to the COVID-19 Pandemic and Table 2. Cost of Doing Nothing: Estimated Societal Costs from Indirect Health Harms of the 211 COVID-19 Pandemic - should include confidence intervals for estimates, as included in S1.

- The description of the ‘Selection of health condition to be included in BRACE model’ in the main paper is quite brief - with further details provided in S3. However, it is unclear as to why other public health conditions with more available evidence in Table C1 (ex. Heart disease, Suicide/self-harm, Diabetes mellitus) were not included, while housing insecurity was selected. Please provide further details regarding the decision process.

- In Table B1 in Supplementary File 2 - given the strength of evidence was much stronger for the ‘Social determinants of health’ factors related to Food insecurity, compared to housing insecurity and Job Loss, why did the authors select Homelessness as the included Public Health Condition?

- The reach of public awareness campaigns to reduce stroke mortality was assumed at 100% - why? This seems unrealistic. Suggest reducing this based on previous public awareness campaigns.

https://pubmed.ncbi.nlm.nih.gov/23013373/

https://pubmed.ncbi.nlm.nih.gov/33875043/

https://pubmed.ncbi.nlm.nih.gov/34844126/

https://pubmed.ncbi.nlm.nih.gov/9565010/

- Costs for a public awareness campaign ($14,350) appear low if the assumption is 100% coverage of those at risk of stroke. Where did these costs come from? Consider revising based on large public health campaigns.

- It would be helpful to include more detail in Suppl S1 - inputs and results regarding what constitutes the non-medical direct costs.

- In-hospital stroke mortality increased 53% during COVID-19 compared 200 to before - yet, it was unclear as to what was contributing to such a stark increase. Please provide some additional discussion as to the likely causes for such increase

- Given recent data on homelessness was not available due to the eviction moratoria currently in place in California, which is an effective intervention itself, the authors should consider adjusting down the estimate derived from the Great Recession from late 2000s to ensure they do not overestimate the impact of COVID on homelessness.

Reviewer #2: Comments to authors

Thank you for the opportunity to review this manuscript. In this study, the authors estimate the cost-effectiveness of addressing various phenomena which were exacerbated as result of the COVID-19 pandemic in California (depressive symptoms, intimate partner violence, homelessness, excessive alcohol use, opioid use disorder, stroke mortality). The authors find mitigation strategies for each of the six phenomena to be cost saving after a year. I have a few concerns with the study implementation which I detail below.

General comments

For an economic evaluation, this study approach is atypical. I understand including several heterogenous phenomena in one study requires a generalized approach. The authors sufficiently describe this approach in the manuscript. However, I feel a key limitation of this approach is the sacrifice of depth for breadth. Due to data limitations, many strong assumptions are necessary for the authors to implement this model. I detail these concerns below which also include recommendations on improving the methodology.

Specific comments

1. Title: the title/abstract should mention California as this is the perspective of the study.

2. Page 4 lines 84-85: These data sources should be mentioned in the abstract as well.

3. Page 5 lines 88-89: As this is the primary focus of the publication, the BRACE workbook should be uploaded and shared as supplemental information.

4. Page 5 lines 90-97: The model results hinge on assuming all changes in the prevalence of specific phenomena are attributable to COVID. This is a strong assumption and should be reconsidered.

5. Page 5 line 97: Why 20%? This number should be further justified.

6. Table 1: Please add a source column.

7. Page 7 lines 128-136: The assumptions of 20% reach or 100% reach (for stroke) are arbitrary and should be replaced with justifiable estimates supported by the medical literature.

8. Page 8 lines 142-143: The use of the Great Recession as a proxy for homelessness is not appropriate. The economic impacts of the Great Recession and the COVID-19 pandemic are markedly different in that the Great Recession was primarily attributable to the U.S. housing market whereas many jurisdictions implemented eviction moratoria during the COVID-19 pandemic. Alternative data should be used.

9. Page 8 line 149: The reduction in literature-derived health outcomes by 75% is arbitrary and should be replaced with estimates from the literature.

10. Page 9 lines 168-169: These costing details should be provided as a supplement.

11. Page 9 line 176: This 60% figure is arbitrary and should be justified or replaced.

12. Table 2: The estimated societal cost for stroke mortality does not have face validity ($16,773). For example, Wang et al. estimates the cost of hospitalizations for stroke at $20,396 (in 2008 USD), which is $29,422.40 when inflated to 2021 USD using the medical component of the consumer price index. In addition, Wang’s is likely an underestimate as it does not include costs to society.

Reference: Wang et al. J Stroke Cerebrovasc Dis. 2014 ; 23(5): 861–868. doi:10.1016/j.jstrokecerebrovasdis.2013.07.017.

13. Table 3. Why is each intervention compared to do-nothing in a pairwise manner? A proper methodological approach would be to conduct one economic evaluation with all comparators in addition to do-nothing evaluated together. This would involve evaluating dominance, extended dominance, etc. with resulting interventions remaining on the cost-effectiveness frontier assigned ICERs.

14. Page 14. Lines 242-252: The authors provide 95% confidence intervals for the findings from the study. However, generating confidence intervals from probabilistic sensitivity analysis results, i.e., Monte Carlo simulations, is not straightforward. How were these intervals constructed – was bootstrapping or another statistical method used? The methods to generate these bounds should be clearly described in the main text and briefly mentioned in the abstract. Are these bounds simply the 2.5th and 97.5th percentiles of the PSA outputs? If so, these are not valid confidence intervals.

6. PLOS authors have the option to publish the peer review history of their article (what does this mean?). If published, this will include your full peer review and any attached files.

Reviewer #1: No

Reviewer #2: No

---

## [Author Response · Author response to Decision Letter 0]

23 Jun 2022

We thank the reviewers for their thorough review of our manuscript and the detailed comments they have provided. Below, we have responded to each point that was raised.

Reviewer #1: This interesting paper presents data on the cost-effectiveness of key evidence-based mitigation strategies that address six priority health conditions that have been indirectly affected by the COVID-19 pandemic in California. It is well written and results are presented clearly and conclusions are drawn from data presented.

As COVID-19 vaccine population coverage continues to increase in many countries, many governments are keen to ease both health and border restrictions to enable economic rebound and communities to recover under ‘COVID-normal’ policies. However, as experts predicted the COVID-19 restrictions and acute focus on managing the COVID-19 pandemic meant that many other health conditions have been deprioritized in the process and over the next 1-5 years, we are likely to see a second wave of health issues that have resulted from this deprioritization.

The authors undertook a systematic review of the recent literature to identify key health conditions that have been exacerbated during COVID pandemic - and findings of the review and selection of the six conditions are presented in supplementary files.

I only have a few minor suggestions:

- Table 1. The Human Toll: Increased Health Harms due to the COVID-19 Pandemic and Table 2. Cost of Doing Nothing: Estimated Societal Costs from Indirect Health Harms of the 211 COVID-19 Pandemic - should include confidence intervals for estimates, as included in S1.

RESPONSE: Thank you. We now added the uncertainty range for the relative risk values.

- The description of the ‘Selection of health condition to be included in BRACE model’ in the main paper is quite brief - with further details provided in S3. However, it is unclear as to why other public health conditions with more available evidence in Table C1 (ex. Heart disease, Suicide/self-harm, Diabetes mellitus) were not included, while housing insecurity was selected. Please provide further details regarding the decision process.

RESPONSE: Health conditions were selected primarily based on availability of robust data at the time on impact of COVID on the prevalence of the condition. Potential for cascading effects (i.e., ACEs impacts) were also considered. This study was part of a report prepared for the Office of the California Surgeon General, and their priorities were taken into account to narrow down our selection.

- In Table B1 in Supplementary File 2 - given the strength of evidence was much stronger for the ‘Social determinants of health’ factors related to Food insecurity, compared to housing insecurity and Job Loss, why did the authors select Homelessness as the included Public Health Condition?

RESPONSE: Our selection of health conditions was informed by the factors described above. Additionally, the selected conditions needed to be specific enough so that outcomes could be modeled; social determinants of health was too broad. Furthermore, homelessness was selected less for actual observed increase in prevalence than a concern that without eviction moratoria there would be a big jump.

- The reach of public awareness campaigns to reduce stroke mortality was assumed at 100% - why? This seems unrealistic. Suggest reducing this based on previous public awareness campaigns.

https://pubmed.ncbi.nlm.nih.gov/23013373/

https://pubmed.ncbi.nlm.nih.gov/33875043/

https://pubmed.ncbi.nlm.nih.gov/34844126/

https://pubmed.ncbi.nlm.nih.gov/9565010/

RESPONSE: The eligible population to receive this intervention was people at risk for in-hospital stroke mortality (i.e., not everyone who is at risk for stroke) and we assumed 100% potential reach. We now clarify this within the text. We agree that this may not be feasible, and we now added a one-way sensitivity analysis on this parameter in Suppl S1. Briefly, when campaign coverage was halved, all health and cost outcomes were also reduced at scale, and the intervention remained dominant. 

- Costs for a public awareness campaign ($14,350) appear low if the assumption is 100% coverage of those at risk of stroke. Where did these costs come from? Consider revising based on large public health campaigns.

RESPONSE: This cost per person was derived from peer-reviewed literature, adjusted for the California population and proportion at risk for in-hospital stroke mortality (i.e., those at risk for delayed care for stroke), as described above. 

- It would be helpful to include more detail in Suppl S1 - inputs and results regarding what constitutes the non-medical direct costs.

RESPONSE: The exact composition of non-medical direct costs differs for each condition, partly because each condition has a unique set of associated costs, and also because these data came from different sources. Each study used a different set of cost items, and we selected the data points that were the most comprehensive. We now added to Suppl S1 brief summaries of what each source included when calculating costs.

- In-hospital stroke mortality increased 53% during COVID-19 compared to before - yet, it was unclear as to what was contributing to such a stark increase. Please provide some additional discussion as to the likely causes for such increase

RESPONSE: Thank you, we now briefly discuss why stroke mortality may have increased (lines 283-286). 

- Given recent data on homelessness was not available due to the eviction moratoria currently in place in California, which is an effective intervention itself, the authors should consider adjusting down the estimate derived from the Great Recession from late 2000s to ensure they do not overestimate the impact of COVID on homelessness.

RESPONSE: We agree that an eviction moratorium was an effective intervention. However, eviction moratoria in California were issued temporarily, for defined periods of time, and there was uncertainty about whether they would be extended. The large impact on livelihoods suggested that homelessness might have increased once the moratoria were lifted. Figures A5 and A6 in Suppl S1 include 1-way sensitivity analyses on this relative risk: net savings are still achieved with lower relative risks and substantial QALYs are gained. We now added to Suppl S1 an additional scenario where the relative risk is halved.

Reviewer #2: Comments to authors

Thank you for the opportunity to review this manuscript. In this study, the authors estimate the cost-effectiveness of addressing various phenomena which were exacerbated as result of the COVID-19 pandemic in California (depressive symptoms, intimate partner violence, homelessness, excessive alcohol use, opioid use disorder, stroke mortality). The authors find mitigation strategies for each of the six phenomena to be cost saving after a year. I have a few concerns with the study implementation which I detail below.

General comments

For an economic evaluation, this study approach is atypical. I understand including several heterogenous phenomena in one study requires a generalized approach. The authors sufficiently describe this approach in the manuscript. However, I feel a key limitation of this approach is the sacrifice of depth for breadth. Due to data limitations, many strong assumptions are necessary for the authors to implement this model. I detail these concerns below which also include recommendations on improving the methodology.

Specific comments

1. Title: the title/abstract should mention California as this is the perspective of the study.

RESPONSE: Thank you. Our abstract already mentions in the methods paragraph that we modeled outcomes for California (line 28-30).

2. Page 4 lines 84-85: These data sources should be mentioned in the abstract as well.

RESPONSE: Thank you. We have added this to the abstract. 

3. Page 5 lines 88-89: As this is the primary focus of the publication, the BRACE workbook should be uploaded and shared as supplemental information.

RESPONSE: We provide abundant detail in the supplements about out methods, including all data sources, assumptions, and calculations. We can provide the interactive workbook to readers upon request. 

4. Page 5 lines 90-97: The model results hinge on assuming all changes in the prevalence of specific phenomena are attributable to COVID. This is a strong assumption and should be reconsidered.

RESPONSE: Our “change during COVID” data come from peer-reviewed articles that were recently published at the time of the analysis and showed statistically significant changes in the conditions during and immediately before COVID (with the exception of homelessness). These changes can be due to COVID itself, because of public health responses to COVID, or due to unrelated reasons. We do not aim to distinguish between the cause of increased prevalence. It is obvious and (now) very well established that COVID and the public health response has led to significant disruptions in people’s lives, so it is plausible (and very likely) that the pandemic and the response may be the underlying reasons for these changes. While it is true that certain individuals might have experienced these conditions unrelated to COVID-19, the pandemic remains the most plausible explanation for these large population-level changes in prevalence and severity of these downstream health conditions. 

Additionally, as we explain in lines 133-135, we assume that interventions would be implemented in eligible populations (i.e., those with the condition) regardless of whether the condition is caused by the COVID pandemic. For example, people with pre-existing or non-COVID-related recent onset depression are also eligible to receive CBT and SSRI. Indeed, it may be impossible to distinguish. The magnitude of cost-savings and health benefits are determined based on the amount of change that occurred in the prevalence of the depression, regardless of why that changed occurred. 

5. Page 5 line 97: Why 20%? This number should be further justified.

RESPONSE: It is an assumption intended to be ambitious but realistic. Greater or lesser coverage would increase or decrease the magnitude of the findings to scale, but this would not alter the nature and interpretation of findings. 

6. Table 1: Please add a source column.

RESPONSE: We now added references to Table 1 and Table 2.

7. Page 7 lines 128-136: The assumptions of 20% reach or 100% reach (for stroke) are arbitrary and should be replaced with justifiable estimates supported by the medical literature.

RESPONSE: We now note the treatment coverage levels for several of the studied conditions (lines 136-137). This serves as a useful point of reference, but the incremental coverage is largely a public health policy decision, although constrained by available capacity and resources. We now emphasize that smaller efforts would result in smaller gains, but identical patterns of net savings (lines 371-373). 

8. Page 8 lines 142-143: The use of the Great Recession as a proxy for homelessness is not appropriate. The economic impacts of the Great Recession and the COVID-19 pandemic are markedly different in that the Great Recession was primarily attributable to the U.S. housing market whereas many jurisdictions implemented eviction moratoria during the COVID-19 pandemic. Alternative data should be used.

RESPONSE: We agree that the two events have led to different outcomes. This is why we adjusted the health and cost effects associated with homelessness, assuming 75% lower health consequences. We used similar rates of potential homelessness because of the very steep jump in unemployment but as noted above, a smaller scale would affect the estimated magnitude but not the nature of the net economic benefits. We now report a wide sensitivity analysis (lines 261-265).

9. Page 8 line 149: The reduction in literature-derived health outcomes by 75% is arbitrary and should be replaced with estimates from the literature.

RESPONSE: Given eviction moratoria and the recency of the pandemic, these expected health outcomes have not yet occurred. As such, there is no direct evidence on the mortality associated with COVID-19-related homelessness, which is why we used a very conservative assumption. 

10. Page 9 lines 168-169: These costing details should be provided as a supplement.

RESPONSE: Thank you. All our data sources are listed in Suppl 1. We now also note in Suppl 1 tables what types of costs each cost input included.

11. Page 9 line 176: This 60% figure is arbitrary and should be justified or replaced.

RESPONSE: We agree that this was an estimate. We now include sensitivity analyses in Suppl 1.

12. Table 2: The estimated societal cost for stroke mortality does not have face validity ($16,773). For example, Wang et al. estimates the cost of hospitalizations for stroke at $20,396 (in 2008 USD), which is $29,422.40 when inflated to 2021 USD using the medical component of the consumer price index. In addition, Wang’s is likely an underestimate as it does not include costs to society.

Reference: Wang et al. J Stroke Cerebrovasc Dis. 2014 ; 23(5): 861–868. doi:10.1016/j.jstrokecerebrovasdis.2013.07.017.

RESPONSE: Thank you. The value $16,773 is the mean of one-time costs for three different types of strokes, which represent the medical direct cost, obtained from the Healthcare Cost and Utilization Project (HCUP). HCUP includes public and private insurances.

Wang et al "excluded patients with capitated plans because their costs of hospitalization would not reflect all the medical services provided to them." They also excluded all patients above age 64. The inclusion of only employer-sponsored insurance, exclusion of Medicare and capitation plans likely led to Wang et al.’s higher estimates. Additionally, while Wang et al. aimed to capture an estimated cost that reflects all medical services provided to patients, the cost they use is the payment amount to providers and only payment amount from private insurances -- which may be skewed to the higher end. We believe our estimate of $16,773 is more representative of the population.

Regardless, greater cost would lead to more potential savings by preventing stroke mortality; the intervention would remain dominant and would likely be even more desirable to policymakers given greater net savings. 

13. Table 3. Why is each intervention compared to do-nothing in a pairwise manner? A proper methodological approach would be to conduct one economic evaluation with all comparators in addition to do-nothing evaluated together. This would involve evaluating dominance, extended dominance, etc. with resulting interventions remaining on the cost-effectiveness frontier assigned ICERs.

RESPONSE: We ordinarily agree with the proposed approach, but we did not envision this as an integrated decision problem for the state policy makers. Instead, a series of independent decisions. 

We did not intend to investigate which health condition would be most cost-effective to intervene in. Our goal was to demonstrate the large health harms and costs associated with the rise in each condition, and that there is a possibility to avert these outcomes. We attempted to provide a menu of reasonable interventions that could alleviate suffering and costs that would provide an evidence-base for policy makers in California, but we do not believe it would be appropriate to compare such vastly different health conditions to one another. 

We note that with net savings for all the interventions over time, the economically optimal choice for the state would probably be to implement all of them (i.e., there are no ICERs over several years).

14. Page 14. Lines 242-252: The authors provide 95% confidence intervals for the findings from the study. However, generating confidence intervals from probabilistic sensitivity analysis results, i.e., Monte Carlo simulations, is not straightforward. How were these intervals constructed – was bootstrapping or another statistical method used? The methods to generate these bounds should be clearly described in the main text and briefly mentioned in the abstract. Are these bounds simply the 2.5th and 97.5th percentiles of the PSA outputs? If so, these are not valid confidence intervals.

RESPONSE: Yes, they are 2.5 and 97.5 percentiles of PSA outputs, which represent the range of possible outcomes given simultaneous variation of all inputs, given specified confidence bounds and assumption of independence between inputs. Because the term CI invokes empirical statistics, we now revised our manuscript to use the term “prediction interval (PI)”.

---

## [Editor Report · Decision Letter 1]

5 Jul 2022

Indirect COVID-19 Health Effects and Potential Mitigating Interventions: Cost-effectiveness framework

PONE-D-22-09397R1

Dear Dr. Sigal Maya,

We’re pleased to inform you that your manuscript has been judged scientifically suitable for publication and will be formally accepted for publication once it meets all outstanding technical requirements.

Kind regards,

Sebsibe Tadesse, PhD

Academic Editor

PLOS ONE
---

## [Editor Report · Acceptance letter]

8 Jul 2022

PONE-D-22-09397R1 

Indirect COVID-19 Health Effects and Potential Mitigating Interventions: Cost-effectiveness framework 

Dear Dr. Maya:

I'm pleased to inform you that your manuscript has been deemed suitable for publication in PLOS ONE. Congratulations! Your manuscript is now with our production department. 

Kind regards, 

on behalf of

Dr. Sebsibe Tadesse 

Academic Editor

PLOS ONE